# Approximating Family of Steep Traveling Wave Solutions to Fisher's Equation with PINNs

**Franz M. Rohrhofer & Bernhard C. Geiger**[*]
Know-Center GmbH
Sandgasse 36, 8010 Graz, Styria, Austria
{frohrhofer, bgeiger}@know-center.at

**Stefan Posch & Clemens Gößnitzer**
LEC GmbH
Inffeldgasse 19, 8010 Graz, Styria, Austria
{stefan.posch, clemens.goessnitzer}@lec.tugraz.at

## Abstract

In this paper, we adapt Physics-Informed Neural Networks (PINNs) to solve Fisher's equation with solutions characterized by steep traveling wave fronts. We introduce a residual weighting scheme that is based on the underlying reaction dynamics and helps in tracking the propagating wave fronts. Furthermore, we explore a network architecture tailored for solutions in the form of traveling waves. Lastly, we assess the capacity of PINNs to approximate an entire family of traveling wave solutions by incorporating the reaction rate coefficient as an additional input to the network architecture.

## 1 Introduction

Reaction-diffusion systems constitute a wide class of mathematical models used in biology, physics, chemistry, ecology, and engineering. Fisher's equation, a simple yet profound reaction-diffusion system, was introduced in (Fisher, 1937) and reads

$$\frac{\partial u}{\partial t} - \frac{\partial^2 u}{\partial x^2} = \rho u(1 - u), \tag{1}$$

where $u$ is the concentration of a chemical substance, and $\rho u(1 - u) \equiv R(u; \rho)$ is a reaction term, parameterized by the reaction rate coefficient $\rho$. This equation admits traveling wave solutions of the form $u(x, t) = u(x \pm ct) \equiv u(z)$ for every wave speed $c \geq 2\sqrt{\rho}$. For values $\rho \gg 1$, the solution exhibits a sharp and fast traveling wave front which demands fine spatial and temporal resolutions when studied numerically (Zhao & Wei, 2003).

Physics-informed neural networks (PINNs) have emerged as a deep learning method that is applicable to physical systems such as described by Eq. (1) (Raissi et al., 2019; Pan et al., 2023). The utility of PINNs in solving differential equations is still under investigation, but their flexibility and the fact that they do not need discretization has already shown great potential for solving all kinds of problems involving differential equations (Raissi et al., 2020). Training PINNs – i.e., obtaining the neural approximation $u_\theta$ of the solution function $u$ in Eq. (1) by minimizing residuals on collocation points – however, is not straightforward and demands a carefully chosen optimization procedure.

In this work, we apply PINNs to approximate solutions to Fisher's equation with large reaction rate coefficients $\rho$. We observe that standard PINNs struggle in approximating the resulting sharp solutions, and introduce a novel residual weighting method as a remedy. Additionally, we assess the effectiveness of a specific network architecture, designed to automatically adapt to the shape of traveling wave fronts. Finally, we investigate the generalization capability of PINNs by employing a single PINN to approximate a family of traveling wave solutions parameterized by the reaction rate coefficient $\rho$, where said coefficient is an additional input to the PINN.

---

[*]Corresponding author: Franz M. Rohrhofer

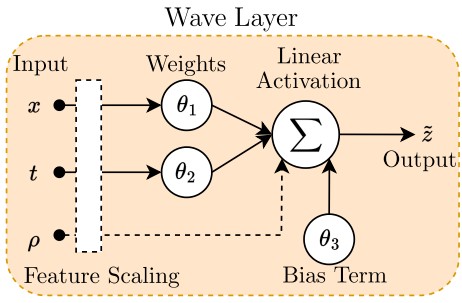
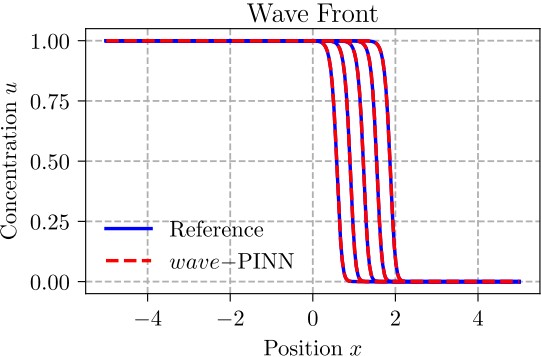

Figure 1: (*Left*) Wave layer structure compelling the network function to $u(\tilde{z})$ with $\tilde{z}(x,t) = \theta_1 x + \theta_2 t + \theta_3$ for the discrete-$\rho$ approximation, or $\tilde{z}(x,t;\rho) = \theta_1 \rho_1 x + \theta_2 \rho_2 t + \theta_3$ for the continuous-$\rho$ approximation. (*Right*) Reference and predicted wave front for $\rho = 10{,}000$.

## 2 METHODOLOGY

**Steep Traveling Wave Solutions.** For Eq. (1), the traveling wave solution with a constant wave speed of $c = 5\sqrt{\rho/6}$ is of particular interest, as it admits the analytical solution (Ablowitz & Zeppetella, 1979):

$$u(x,t;\rho) = \left[1 + \exp\left(\sqrt{\frac{\rho}{6}}x - \frac{5\rho}{6}t\right)\right]^{-2} \tag{2}$$

that satisfies $\lim_{z \to -\infty} u(z) = 0$ and $\lim_{z \to \infty} u(z) = 1$. Although Eq. (1) and Eq. (2) can be brought into a nondimensionalized form, e.g. by using $\hat{x} \leftarrow \sqrt{\rho}x$ and $\hat{t} \leftarrow \rho t$, the system is numerically studied often with the dependency on the reaction rate coefficient $\rho$. This is due to the fact that for a fixed computational domain, the reaction term, and thus the steepness of the traveling wave front, can be made arbitrarily large by adjusting $\rho$. Typical values fall into the range $\rho \in (1{,}000, 10{,}000)$ and these settings commonly serve as a numerical challenge for discretization schemes (Li et al., 1998; Qiu & Sloan, 1998; Kırlı & Irk, 2023).

**Residual Weighting.** PINNs are known to suffer from optimization issues when applied to problems with sharp and shock-like solutions (Liu et al., 2024; Mao & Meng, 2023). Therefore, with $u_\theta$ being the neural approximation corresponding to network parameters $\theta$, we propose a residual weighting scheme that is based on the reaction dynamics $R$ and assigns the weight

$$\omega = \frac{1}{\lambda|R(u_\theta; \rho)| + 1} \tag{3}$$

to each residual $f$ in the physics loss function $\mathcal{L}(\theta) \sim \sum_i |\omega_i f_i|^2$ (cf. Eq. (3) in (Raissi et al., 2019)). Intuitively, for $\lambda > 0$, the weights $\omega$ reduce the influence of steep gradients in regions with predominant reaction dynamics, while leaving residuals at predominant diffusion dynamics unaffected. Our approch is thus similar in spirit to (Liu et al., 2024) and in contrast to approaches that place additional residual points in regions of high residuals with (Mao & Meng, 2023) or that weight collocation points with high residuals more strongly (McClenny & Braga-Neto, 2023).

**Standard- & Wave-Architecture.** In the *standard* architecture, hidden layers directly follow the input layer, aiming to approximate the solution function in the general spatial-temporal form $u(x,t)$. Conversely, in the *wave* architecture, an additional wave layer is introduced that directly follows the input layer and precedes the hidden layers. The wave layer introduces a latent variable representation (cf. Figure 1) to compel the network function to solutions that take the traveling wave form $u(\tilde{z})$. This approach is similar in spirit to the Lagrangian PINN (Mojgani et al., 2023) and the characteristics-informed neural network (Braga-Neto, 2022), which both adapt the network architecture to the solution family of the respective (class of) PDEs.

Table 1: The mean (and standard deviations) of the $L_2$-error for the four tested models and discrete values of $\rho$. All numbers should be multiplied by $10^{-4}$. **Bold:** Best results.

| Model | $\lambda$ | Reaction rate coefficient, $\rho$ | | |
| | | 100 | 1,000 | 10,000 |
|---|---|---|---|---|
| *standard*-ANN | - | 1.66 (1.30) | 0.79 (0.29) | 1.69 (2.10) |
| *wave*-ANN | - | 1.12 (0.60) | 1.05 (0.81) | 1.54 (1.62) |
| *standard*-PINN | 0 | 180 (46) | 320 (184) | 5689 (3469) |
| | 0.1 | 37.9 (11.0) | 81.1 (15.9) | 36.8 (34.3) |
| | 1 | 5.74 (0.95) | 14.0 (3.0) | 132 (66) |
| | 10 | 1.72 (1.23) | 8.79 (1.46) | 354 (174) |
| *wave*-PINN | 0 | 16.5 (13.5) | 178 (160) | 5082 (4106) |
| | 0.1 | 1.26 (1.49) | 1.65 (0.95) | 12.1 (6.5) |
| | 1 | **0.82 (0.95)** | **0.60 (0.63)** | **1.10 (0.80)** |
| | 10 | 2.16 (3.28) | 0.83 (1.45) | 3.11 (3.71) |

## 3 RESULTS

For our experiments, we consider the following two applications: the discrete-$\rho$ approximation (i.e., approximating $u(x, t)$ for a single $\rho$), and the continuous-$\rho$ approximation (i.e., approximating an entire family of solutions $u(x, t; \rho)$) by applying a *generalizing* architecture that takes $\rho$ as an additional input. For each application, we assess the $L_2$-error of four specific models: (i) *standard*-ANN and (ii) *wave*-ANN, which are purely data-driven models trained on the solution function (2), by sampling labeled data from it anew at each epoch; and (iii) *standard*-PINN and (iv) *wave*-PINN, which infer the solution function by minimizing the physics loss and a data loss that encodes the initial and boundary conditions.[1]

We consider $x \in [-5, 5]$ as the spatial and $t \in [0, 0.004]$ as the temporal domain and vary reaction rate coefficients in the range $\rho = [100, 10{,}000]$. In the *standard* architecture, all input variables are scaled to $[0, 1]$. Conversely, in the *wave* architecture, only the temporal variable is scaled to $[0, 1]$ to ensure reduced sensitivity of the $\theta_j$ weights in the wave layer. The reaction rate input $\rho$ is further transformed to $\rho_1 \leftarrow \sqrt{\rho}$ and $\rho_2 = \rho$ (cf. Figure 1).

Datasets of size $N = 1024$ are sampled anew at each training epoch using Latin hypercube sampling; the test set is taken with $N = 10{,}000$. The PINN for a single value of $\rho$ and the *generalizing* architecture have two and three hidden layers, respectively, with 20 neurons each. We use the hyperbolic tangent (tanh) as hidden layer activation and the Sigmoid activation as output layer activation to ensure that any prediction lies in the bounded interval $u_\theta \in [0, 1]$. Optimization of all models is carried out with Adam for $50k$ training epochs for the discrete-$\rho$ approximation and for $100k$ epochs when applying the *generalizing* architecture. A default learning rate of $0.001$ is applied.

### 3.1 DISCRETE-$\rho$ APPROXIMATION

For the discrete-$\rho$ experiment, the model performance is evaluated by approximating the solution function for three individual reaction rate coefficients $\rho \in \{100, 1{,}000, 10{,}000\}$ while applying separate models for each case. The result can be found in Table 1 which shows the mean (and standard deviations) of the $L_2$-error over ten independently trained model instances.

As evident from the table, the *wave*-PINN with $\lambda = 1$ achieves the best results, even outperforming both data-driven models across all $\rho$ values. This is particularly intriguing, as the data-driven ANNs have effectively (and without overfitting) learned the solution from labeled data, making them highly accurate for the given network size. Furthermore, the right panel of Figure 1 demonstrates that the *wave*-PINN with $\lambda = 1$ accurately captures the steep wave front at $\rho = 10{,}000$.

By comparing the *standard* and *wave* architecture, we observe that the *wave* models generally outperform the *standard* models, with a notable discrepancy observed within the PINN models. Across

---

[1]Initial and boundary conditions are imposed by sampling labeled data from the analytical solution at the computational boundary.

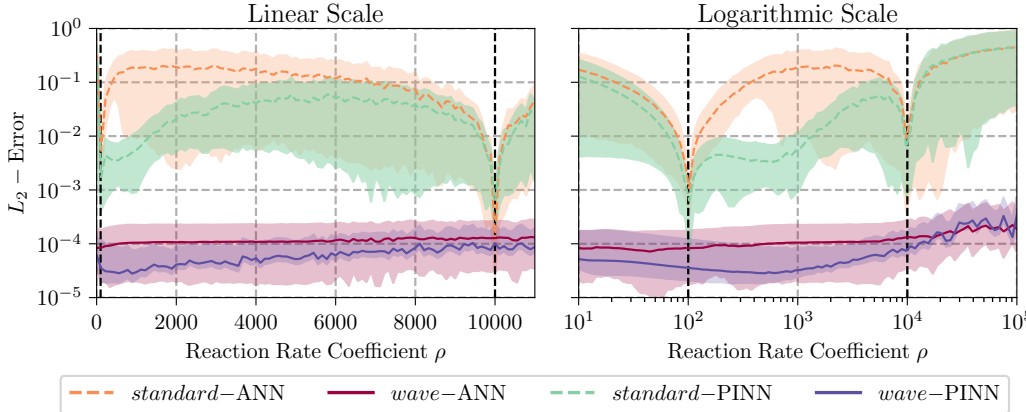

Figure 2: The median (and 25%- and 75%-quantiles) of the $L_2$-error for each tested model on the continuous domain $\rho \in [100, 10{,}000]$. Labeled training data was used at $\rho = 100$ and $\rho = 10{,}000$, indicated by the dashed black lines.

nearly all residual settings ($\lambda$), the *wave*-PINN consistently achieves lower prediction errors compared to the *standard*-PINN, while the difference is less pronounced for the data-driven ANNs.

Regarding the optimal residual weighting, we find that a value of $\lambda = 1$ is effective for the *wave*-PINN, whereas for the *standard*-PINN, the optimal choice is less evident. Nevertheless, the results clearly indicate that conventional, unweighted ($\lambda = 0$) PINNs exhibit significant performance issues, particularly for large $\rho$ values. Across all values of $\rho$, employing a residual weight with $\lambda > 0$ yields considerably better performance compared to $\lambda = 0$.

## 3.2 CONTINUOUS-$\rho$ APPROXIMATION

For the continuous-$\rho$ experiment, we utilize labeled data sampled from the analytical solution for $\rho_{\min} = 100$ and $\rho_{\max} = 10{,}000$ and aimed for obtaining a reliable approximation within the continuous range $\rho \in [\rho_{\min}, \rho_{\max}]$. While collocation points for the PINN instances are sampled within this range, the data-driven models are exclusively trained on labeled data corresponding to $\rho_{\min}$ and $\rho_{\max}$. The results for this interpolation task are depicted in Figure 2 which shows the median (and 25%- and 75%-quantiles) of the $L_2$-error across ten independently trained model instances.

Consistent with the results from the previous experiment, the *wave*-PINN achieves the best performance across the continuous $\rho$ domain. However, we also observe remarkable performance from the *wave*-ANN, considering that it is trained solely on the solution at the two specific $\rho$ values (indicated by dashed black lines in Figure 2). The inclusion of the wave layer appears to offer a highly effective inductive bias that facilitates finding solutions over a continuous parameterization. Notably, even within the range of $\rho = 100{,}000$, both *wave* models continue to yield strong approximations in the extrapolation regime, with only a slight degradation compared to the interpolation regime.

Overall, we see that the PINN models outperform their data-driven counterparts and that the *wave* architectures outperform the *standard* architectures. Apparently, the inductive bias induced by the wave layer is stronger than the learning bias induced by the physics loss function. This is expected, as the wave layer, especially in the generalizing architecture, incorporates explicit knowledge about traveling wave nature of the solution function.

## 4 CONCLUSION

In this paper, PINNs were applied to solve Fisher's equation with large reaction rate coefficients and sharp solutions representing steep traveling wave fronts. To address the challenges posed by sharp transitions in traveling waves, a residual weighting scheme was introduced and integrated into the optimization process of PINNs. The residual weighting scheme is based on the underlying reaction

term and effectively facilitates optimization in sharp regions of the solution function. Additionally, a specific network architecture designed to approximate traveling waves was tested, and the results demonstrated outstanding improvements compared to conventional network architectures. Finally, a generalized PINN architecture was explored that incorporates the reaction rate coefficient as an additional input, enabling the approximation of an entire family of solutions to Fisher's equation. The results showed that the PINN with the introduced modifications can effectively solve Fisher's equation with large reaction rate coefficients and, furthermore, can be extended to approximate a family of sharp solutions with a single PINN instance. Furthermore, our modifications complement and are compatible with other approaches to improving PINN training, such as methods for collocation point sampling (Mao & Meng, 2023) and domain decomposition (Jagtap & Karniadakis, 2020). Future work shall evaluate the utility of our work in the context of this literature.

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
