# OpenReview forum: "Approximating Family of Steep Traveling Wave Solutions to Fisher's Equation with PINNs"
_ICLR.cc/2024/Workshop/AI4DiffEqtnsInSci — AI4DiffEqtnsInSci @ ICLR 2024 Poster_

### Official Review · Reviewer_1zSu · 2024-02-24
**Approximating Family of Steep Traveling Wave Solutions to Fisher's Equation with PINNs**

**Rating:** 6
**Confidence:** 3

**Review:**

## General comments
The manuscript titled 'Approximating Family of Steep Traveling Wave Solutions to
Fisher’s Equation with PINNs' presents an innovative approach using Physics-
Informed Neural Networks to model steep traveling wave solutions in Fisher’s
equation. The proposed method, including a novel residual weighting technique
and a specialized network architecture, demonstrates improved accuracy in
approximating sharp solution profiles.
The findings are promising. I therefore recommend that the paper be published but
after the authors address the comments presented below, which are mostly minor.

## Comments
1. Here the efficacy of the proposed method is compared with traditional PINN
only. There are multiple variants of PINN [1,2,3,4, etc.]. The authors are
encouraged to spend a few words on better contextualizing their work
within the broader context of the current body of literature.
2. The values of the reaction rate coefficients are written as 1.000, 10.000. I am
not sure if they mean 1 and 10. However, if the authors mean 1,000 or 10,000,
those should be revised.
3. “Optimization is carried out with Adam for 50k training epochs for the
discrete-ρ approximation and for 100k epochs when applying the
generalizing architecture. A default learning rate of 0.001 is applied.”
Authors should clarify if these are true for both PINN and wave-PINN ?
4. The authors are advised to add a brief discussion on the limitations of the
method and the NN architecture.

## References:
[1] L. D. McClenny and Ulisses Braga-Neto, “Self-adaptive physics-informed neural
networks,” Journal of Computational Physics, vol. 474, pp. 111722–111722, Feb. 2023.
[2] U. Braga-Neto, “Characteristics-Informed Neural Networks for Forward and Inverse Hyperbolic
Problems,” arXiv.org, 2022.
[3] R. Mojgani, M. Balajewicz, and P. Hassanzadeh, Lagrangian pinns: A causality-conforming
solution to failure modes of physics-informed neural networks, arXiv preprint arXiv:2205.02902,
2022.
[4] A. D. Jagtap and G. E. Karniadakis, Extended physics-informed neural networks (xpinns): A
generalized space-time domain decomposition based deep learning framework for nonlinear partial
differential equations, Communications in Computational Physics, vol. 28, no. 5, pp. 20022041, 2020.

---

### Official Review · Reviewer_i9wC · 2024-02-24
**The authors demonstrate the use of a weighting scheme to balance physics loss term and data loss. They also propose a wave layer that approximates wave form u(z) to guide the neural network for capturing the structure of wave. The proposed solution significantly outperform neural networks without standard ANN and PINN, contributing to solve the target PDE when solution is characterized with steep wave fronts.**

**Rating:** 9
**Confidence:** 5

**Review:**

The paper is well written, particularly in its coverage of background, motivation, and experimental details. The experiments are thorough and well-detailed. However, it is unclear in the explanation of the entire loss function, which is not explicitly outlined.

While the idea of weighting and/or adjusting the residual term is not novel, the proposed approach shows its simplicity and effectiveness. However, it remains unclear whether the proposed method outperforms the effectiveness of the approach discussed in the referenced paper by Liu et al. (2024), which introduced gradient-dependent weights to capture solutions of PDEs. A clearer comparison with existing methods would strengthen the originality aspect of the paper.

The promising results observed in both discrete and continuous experiments highlight the potential effectiveness of the proposed solution. Particularly the proposed weight scheme is effective especially in scenarios where the reaction rate coefficient is smaller. However, the authors fall short in providing a benchmarking case that demonstrates the importance of solving the addressed problem using the proposed solution compared to other existing PINN approaches. Clarifying the extent to which the weighting scheme enhances PINN capabilities would help the significance of the contribution.

---

### Meta-Review · Area_Chair_Pr2G · 2024-03-01

**Recommendation:** Accept (Poster)

**Metareview:**

I would like to thank reviewers for their careful review. Both reviewers seem to agree on accepting the paper. I also went through it and the study of how to enhance PINNs model for capturing sharp moving interfaces (shack) is quite interesting and valuable. I just ask authors to clearly address reviewers' comments in their final paper.  Also, I would like to ask authors to review their references and make sure they cite related papers, I have seen more papers on wave propagation using PINNs, please review and acknowledge those works too.

---

### Decision · Program_Chairs · 2024-03-02

Accept (Poster)